# Effects of the Prescription of Physical Exercises Mediated by Mobile Applications on the Health of Older Adults: A Systematic Review

**DOI:** 10.3390/geriatrics10050122

**Published:** 2025-09-10

**Authors:** Débora Vanessa Santos Dias Costa, Evellin Pereira Dourado, Mayara Bocchi Fernandes, Eduardo Vignoto Fernandes, David Michel de Oliveira

**Affiliations:** 1Department of Physical Therapy, Federal University of Jataí, Jataí 75801-615, GO, Brazil; deboravandias@gmail.com (D.V.S.D.C.); evellinpereiradourado@ufj.edu.br (E.P.D.); 2Department of Medicine, Federal University of Jataí, Jataí 75801-615, GO, Brazil; mayara.fernandes@ufj.edu.br; 3Postgraduate Program in Biosciences and One Health and Postgraduate Program in Health Applied Sciences, Federal University of Jataí (UFJ), Jataí 75801-615, GO, Brazil; eduardovignoto@ufj.edu.br

**Keywords:** health of the elderly, health technology assessment, mobile apps, physical activity, functional health

## Abstract

Background/Objectives: Aging and a sedentary lifestyle aggravate hypokinetic diseases, compromising the functional capacity of older adults. Thus, the prescription of physical exercise (PE) through mobile applications (MA) has emerged as a remote and personalized alternative. However, there are still gaps in the effectiveness of prescribing physical exercise via mobile apps for older people. This study aimed to analyze the effects of prescribing PE through MAs on the health of older adults. Materials and Methods: This systematic review included studies with older people (≥60 years) that used MAs to prescribe PE, published between 2014 and 2024, in Portuguese or English. The search strategy used the descriptors “older adults,” “physical exercise,” “mobile applications,” and “health,” combined with Boolean operators. The screening followed previously defined eligibility criteria regarding population, intervention, outcomes, and study design. Two independent reviewers extracted data, mediated by a third party in case of disagreement; they screened and extracted data from the PubMed and VHL/Medline databases from 2004 to 2024. Risk of bias was assessed according to levels of evidence, and the results were categorized. Results: Of the 2298 publications initially identified, 7 studies were eligible for this review, totaling 748 participants, predominantly female. The studies included prospective and observational clinical trials with older people suffering from Parkinson’s disease, cardiovascular disease, sarcopenia, and breast cancer. The findings showed favorable effects on adherence to the program (6 studies; *n* = 654), an increase in PE (5 studies; *n* = 502), and improvements in functional capacity (4 studies; *n* = 389), perceived quality of life (5 studies; *n* = 481), and muscle strength (3 studies; *n* = 298). Conclusions: The prescription of MA-mediated PE showed positive effects on the health of older people, indicating its viability as a complementary strategy in clinical practice or public health.

## 1. Introduction

The older population has increased globally, including in the Brazilian population, which represents the fifth most populous country in the world [1], and with an even higher proportion in some European countries [2]. With longevity comes an increase in the prevalence of diseases associated with aging, which present considerable challenges for public health [3]. A sedentary lifestyle, a common condition among older adults, is strongly related to the development of hypokinetic comorbidities, higher hospitalization rates, and increased mortality [4]. In addition, reduced mobility contributes to the progressive loss of functional capacity and reduced activities of daily living [5,6]. In this context, the literature points to the regular practice of physical exercise (PE) as essential for increasing muscle strength, cardiovascular function, and behavioral well-being for older people, resulting in greater functional autonomy, a reduction in depressive symptoms, and the prevention of chronic non-communicable diseases [7]. PE adherence is often hindered by psychosocial barriers, such as a lack of company, financial limitations, shortage of time, and the presence of comorbidities, as well as insecurity in performing exercises without supervision [8].

To overcome these barriers, the use of digital technologies, such as mobile applications (MAs), has emerged as an alternative strategy, offering personalized exercise programs adapted to the health conditions of older adults and allowing remote monitoring. This type of intervention has promoted improvements in functional performance, as well as increasing adherence and ensuring greater safety for regular practice [9,10]. In addition, the prescription of PE mediated by MA offers other advantages, such as easier access to health information, reduced costs for travel or face-to-face services, the promotion of digital inclusion, and the flexibility to carry out activities at any time and place, in a practical and autonomous way [11].

However, despite the existing findings, the prescription of physical exercise (PE) mediated by mobile applications (MA) for PE is still little explored, making it essential to synthesize the existing evidence. This information will provide subsidies for the development of effective strategies and for the implementation of PE programs aimed at the older population, both in clinical practice and in public health.

Thus, this study aimed to analyze the effects of prescribing MA-mediated physical exercise on older adults’ health indicators, compared to practicing without technological mediation or intervention.

This study hypothesized that the use of AM favors adherence and promotes positive effects on older adults’ health indicators, making it a promising tool to support PA practice for this population.

## 2. Materials and Methods

This systematic review was carried out in accordance with the PRISMA-P guidelines (Preferred Reporting Items for Systematic Review and Meta-Analysis Protocols) and registered in the PROSPERO CRD42024623375 database [12].

To construct the research question, the “PICO” strategy was adopted: Population/patient; Intervention; Comparison/control; Outcomes (outcome) [13]. Population/patients = Older people aged ≥ 60 years of both sexes; Intervention = Prescription of physical exercise using mobile applications; Comparison/Control = Older people practicing physical exercise without using mobile applications; and Outcomes = applicability or effects on health indicators of older adults. After applying the PICO strategy, the question asked was: “What are the effects of prescribing physical exercise using mobile apps on the health of older adults?”.

### 2.1. Eligibility Criteria

We included complete articles available in electronic databases, in Portuguese and English, published in the last 10 years, and involving participants aged 60 or over. The eligible studies included the following designs: observational studies and randomized and non-randomized clinical trials. In addition, they were required to address the applicability and/or effects of prescribing physical exercise through mobile applications in healthy older people or those with any illness or physical limitation.

During the identification process, a total of 2298 records were retrieved from the databases, including 1717 from Medline and 581 from PubMed. After removing 162 duplicate records, 2136 articles remained for screening.

In the screening phase, 1580 articles were excluded based on titles and abstracts, leaving 556 articles for full-text assessment. No reports were lost in the retrieval process. In the eligibility phase, the 556 full-text articles were assessed, of which 549 were excluded for not meeting all the inclusion criteria (Figure 1).

Of the 549 articles excluded in the eligibility phase, 37 were systematic reviews or meta-analyses that did not specifically address physical activity prescription mediated by mobile applications, avoiding overlapping results. Another 184 studies were excluded because they were outside the thematic scope, 162 because they did not use MA for exercise prescription, and 91 because they only measured physical activity (e.g., number of steps or calories) without evaluating structured exercise prescription. In addition, 75 studies included populations from other age groups, whereas the objective was restricted to older adults (≥60 years). Monographs, theses, and dissertations were also disregarded, as they constitute gray literature.

### 2.2. Database and Search Strategy

The following databases were used for the searches: Medline/Virtual Health Library (VHL) and PubMed (National Library of Medicine). The databases were consulted between March and June 2024. The following descriptors in Portuguese and English, tested in the Health Sciences Descriptors (Decs) and Medical Subject Headings (MeSH), were used as a search strategy: Exercício/Exercise; Aplicativos móveis/Mobile Applications; Idosos/elderly. The descriptors were combined with Boolean operators AND to search for publications and OR to search for similar terms, such as: “Exercise OR Exercise AND Mobile Applications OR Mobile Applications AND Elderly OR elderly OR aged” in Medline/BVS and “Exercise AND Mobile Applications AND elderly” in PubMed.

### 2.3. Data Selection and Collection Process

Phase 1—The studies were identified in the databases and exported to the Zotero^®^ reference manager to exclude duplicates.

Phase 2—The titles and abstracts were read to exclude studies that did not meet the eligibility criteria.

Phase 3—The studies were read in full, excluding the eligibility criteria.

Phase 4—Finally, the data were extracted.

Initially, the titles and abstracts were analyzed to apply the eligibility criteria, followed by full reading of the potentially relevant articles. In the event of disagreement between the evaluators, a third reviewer was consulted for the final decision, ensuring greater methodological rigor and minimizing the risk of bias in the study selection process.

### 2.4. Assessment of the Risk of Bias in Studies

The studies were collected independently by two researchers, and any discrepancies were resolved by a third reviewer. The methodological quality of the studies was classified according to the levels of evidence of the Joanna Briggs Institute [14].

### 2.5. Summary of Results

The extracted data were organized into tables and described narratively, considering the outcomes of interest defined in the review protocol. A descriptive synthesis was performed, grouping the studies according to the observed effects on exercise adherence, functional capacity, quality of life, muscle strength, and other health indicators. The methodological heterogeneity between the studies made it impossible to perform a meta-analysis.

## 3. Results

During the literature review process, 2298 records were initially identified in the two databases: 581 in PubMed and 1717 in BVS/Medline. After removing 162 duplicate records, 2136 articles remained for analysis. In the screening phase, 1580 studies were excluded after analyzing the titles and abstracts because they did not meet the defined criteria. Thus, 556 publications were retrieved for full-text evaluation, resulting in the inclusion of seven articles in the study. Figure 1 below presents a detailed flowchart illustrating the entire article selection and exclusion process.

In addition, the sample was characterized based on the seven articles included, totaling 748 participants who met the general criteria. The sample was predominantly female, due to the health conditions associated with the population studied.

The included studies demonstrated positive effects on several health indicators. There was a significant improvement in adherence to exercise programs [15,16,17,18,19,20], as well as an increase in exercise practice [16,18,21]. Five studies reported gains in perceived quality of life [16,17,18,19,20], while four observed improvements in functional capacity [16,18,19,21], and three reported increased muscle strength [16,17,18]. In addition, two studies highlighted a reduction in depressive symptoms [18,19] and greater acceptance and feasibility of using MAs as a tool to support physical exercise in older adults [15,21].

### Characteristics of the Included Studies

In total, seven studies were included, made up of three randomized clinical trials [15,16,17], one non-randomized clinical trial [18], one prospective single-arm study [19], one prospective pilot study [21], and one cross-sectional observational study [20]. The studies investigated the efficacy of MA in prescribing PE for older people, including populations with chronic diseases such as Parkinson’s, cardiovascular disease, and breast cancer. The main methodological, sample, and outcome characteristics analyzed are described in Table 1 and Table 2.

The studies analyzed were conducted in different regions of the world, including countries in Asia, such as the Republic of Korea [18,19]; Oceania, such as Australia [21]; the European continent, such as Germany and Italy [17,20]; and North America, the United States [15].

Most of the studies were performed in developed countries, with low representation of South American countries.

The selected studies were classified according to the JBI levels of evidence, which use criteria based on the type of study to classify the quality of the articles analyzed (Table 1).

Table 2 presents the characteristics of the studies included—BVS/Medline.

Table 3 outlines the primary features of the studies included from PUBMED, encompassing design, sample characteristics, and key outcomes.

## 4. Discussion

This study aimed to evaluate the effects of prescribing PE through MA on health indicators in older adults. Most of the studies included had a level of evidence of III, with a predominantly female population and characterized by specific health conditions, such as Parkinson’s, musculoskeletal conditions, sarcopenia, myeloid neoplasms, cardiovascular diseases, chronic obstructive pulmonary disease, and breast cancer. The results analyzed point to positive effects in terms of adherence to the programs, improved quality of life, increased functional capacity, and the safety of the interventions.

The study by Ahn et al. (2024) [18] evaluated the use of the Parkinson’s Exercise App, developed by health professionals, in patients with Parkinson’s disease. The sample consisted of 41 participants of both genders, 60% of whom effectively adhered to the two-week home exercise program via smartphone. The protocol included guided videos with moderate intensity exercises aimed at mobility, strength, and coordination. The results showed a significant increase in weekly time spent performing moderate-intensity exercises, as well as a reduction in fatigue, an improvement in mood, and a greater willingness to carry out daily activities. The findings indicate that digital interventions could be a viable alternative for the functional rehabilitation of older people with Parkinson’s, especially when face-to-face access is restricted. However, the results should be interpreted with caution, due to the short intervention time and partial adherence.

The study by Daly et al. (2021) [21] assessed the feasibility of a home-based exercise program aimed at the musculoskeletal health of older people living in the community, using the Physitrack app. Twenty participants performed the intervention for eight weeks, with exercise sessions individualized according to pre-existing clinical conditions. The protocol included muscle strengthening, balance, and mobility exercises, with demonstration videos and clear instructions via the app. The results showed a high program completion rate, few adverse events reported, an increase in self-reported PA, and a high level of satisfaction with the use of technology. These findings suggest that guided digital interventions may be well accepted by older adults, with potential clinical applicability in musculoskeletal prevention and rehabilitation programs. However, the small sample size and short duration of the study limit the generalizability of the results.

The study by Kim et al. (2021) [19] investigated the effectiveness of a personalized MA to manage home exercises in patients with parkinsonism. The protocol lasted eight weeks and included demonstration videos produced by physiotherapists, with stretching, strengthening, balance, and coordination exercises. The 28 participants received individualized prescriptions based on their abilities and preferences. At the end of the study, 87% had completed the program, with increases in the total time and intensity of weekly exercises and adherence to the program, a reduction in depressive symptoms, and an improvement in quality of life. The findings indicate that MAs with a personalized prescription can contribute to the rehabilitation of patients with movement disorders; however, the use of a single-arm design in this study reduces the robustness of the evidence on its effectiveness.

The study by Bonato et al. (2024) [17] evaluated the digital platform GYM (Grow Your Muscle), developed to prescribe home exercises with body weight to older people with sarcopenia. The intervention lasted 48 weeks and was accessed via smartphone, with simple explanatory videos. The results showed an increase in calf strength and good adherence among the participants. The platform stood out for its low cost, ease of use, and accessibility, making it a viable alternative for increasing access to physical rehabilitation for older adults.

The study by Loh et al. (2021) [15] investigated physical exercise intervention for 13 patients with myeloid neoplasms, using the EXCAP^®^ (Exercise for Cancer Patients) program adapted for digital application. The intervention consisted of individualized exercises of mild to moderate intensity, including walking and resistance training with elastic bands, and was accompanied by the PointClickCare application, which recorded data such as exercise frequency and perceived exertion. Although the sample was small, the participants reported that adjusting the intensity facilitated adherence, and the MA helped with monitoring and communication.

In the study by Snoek et al. (2020) [16], the mobile cardiac rehabilitation (MCR) program demonstrated important benefits for patients who avoided conventional rehabilitation, with significant increases in exercise capacity and self-reported physical activity, as well as improvements in VO2 peak after 6 months and 12 months. Diastolic blood pressure and HbA1c levels remained stable in the MCR and increased in the control group. It is therefore understood that the use of MA for physical exercise is considered safe and effective. However, it was difficult to maintain patient adherence. On the other hand, the study conducted by Chung et al. (2024) [22] on self-directed pulmonary rehabilitation using mobile applications for COPD patients showed that mobile applications are challenging tools, due to the variation in adherence to maintain high levels of PE, which requires attention in order to adapt capacity.

The study by Phillips et al. (2017) [20] investigated the preferences of 270 female breast cancer survivors over the age of 60 in relation to technology-supported PE interventions. The majority demonstrated interest in participating in programs remotely, with a focus on muscle strength training exercises, yoga/Pilates^®^, and moderate aerobic activities. However, to increase effectiveness and adherence, interventions should be personalized and include individualized feedback.

In the same period, Phillips et al. (2017) [20] discussed the need for personalized interventions to increase adherence and effectiveness in exercise programs, corroborating the study by Kim et al. (2021) [19]. The study by Soto-Bagaria et al. (2023) [23] searched for MAs aimed at older people to investigate the scientific evidence that supports their use. The authors observed that, out of a total of 11,719 applications, only one stood out: Vivifrail, highlighting the need for personalized exercises to increase adherence and effectiveness. McGarrigle et al. (2020) [24] also point out the importance of recommended and well-founded exercise programs, especially for gaining strength and balance to prevent falls in older adults.

In summary, the included studies demonstrate that MAs are promising tools for prescribing exercise to older adults, with positive effects on the health of this population. However, this review has important limitations: the small number of studies, the short duration of the interventions, the absence of a control group in some trials, unrepresentative samples, and methodological heterogeneity. An additional limitation was the search restricted to two databases (PubMed and VHL/Medline), which, although widely recognized and highly relevant in the health field, may have reduced the scope of the identified literature. In addition, the included studies did not present a standardized plan regarding the type and dosage of the prescribed exercises, which contributed to methodological heterogeneity. The absence of a control group in some of the studies also constitutes a methodological limitation.

To consolidate this evidence, future research using more databases should evaluate the effectiveness of MA in different functional profiles for older adults, with long-term interventions, as well as explore factors such as digital literacy, accessibility, and technological support, which have not yet been addressed but are essential for the safe and effective implementation of these tools.

## 5. Conclusions

The results of the current review indicate that prescribing PE mediated by MA is a viable strategy for promoting the health of older adults, especially to encourage the practice of PE in different clinical conditions common in this population.

The application of these technologies in clinical practice and public health could represent an advance in accessible and personalized care.

## Figures and Tables

**Figure 1 geriatrics-10-00122-f001:**
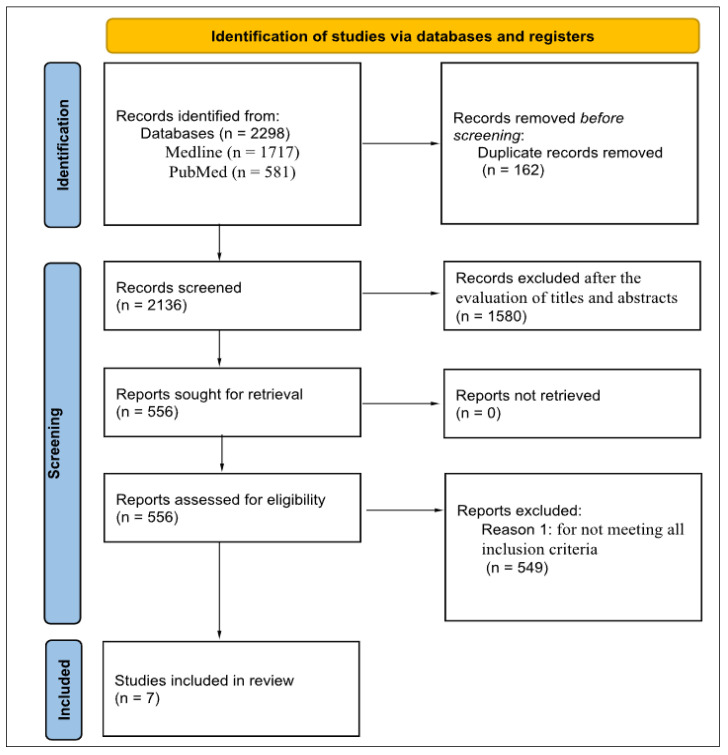
Eligibility of studies included in the review (Prisma Flowchart).

**Table 1 geriatrics-10-00122-t001:** Classification of levels of evidence, according to JBI (2014).

Author and Year	Level of Evidence
Ahn, et al. (2024) [18]	LEVEL III.1
Daly, et al. (2021) [20]	LEVEL III.2
Phillips, et al. (2017) [21]	LEVEL III.2
Loh, et al. (2021) [15]	LEVEL II
Kim, et al. (2021) [19]	LEVEL III.2
Snoek, et al. (2020) [16]	LEVEL II
Bonato, et al. (2024) [17]	LEVEL II

**Table 2 geriatrics-10-00122-t002:** Characteristics of the included studies—BVS/Medline.

Author, Year and Country	Design	Objective	Sample/Methodology	Results	Outcomes
Ahn, et al. (2024) [18]; Republic of Korea	Non-randomized clinical trial	Evaluate the effectiveness of the APP in increasing PA, ↓ cost barriers to exercise, and ↑ QOL.	N = 41, mean age 64.1 years; multicomponent PE with moderate intensity (15 to 20 min with APP: Parkinson Exercise)	↑ Moderate PA, ↑ QOL	25 individuals completed the 2-week program
Daly, et al. (2021) [21]; Australia	Prospective pilot study	Evaluate the functionality and ease of use of the PE prescription APP	N = 20, ≥65 years; multicomponent PE with Physitrack APP (3x/week)	↑ Walking (78 min), ↑ Moderate/vigorous PA (41 min)	Safe and viable for older people monitored by professionals

Legend: APP: Applications; N: Number; PE: Physical Exercise; PA: Physical Activity; QOL: Quality of Life. Symbols: ↑ = increase; ↓ = decrease; ≥: greater than or equal to.

**Table 3 geriatrics-10-00122-t003:** Characteristics of the included studies—PUBMED.

Author, Year and Country	Design	Objective	Sample/Methodology	Results	Outcomes
Phillips et al. (2017) [20]; Germany	Cross-sectional observational	To explore interest in technology-mediated PE programs for breast cancer survivors	N = 279, average age 60.7; Survey on preferences and interests in technology-mediated PE	↑ Advice, ↑ PE remotely	Interventions with technology can be viable and acceptable
Loh et al. (2021) [15]; USA	Randomized clinical trial	Informing the design of an APP-based exercise intervention for patients with myeloid neoplasms	N = 13, mean age 71.6 ± 8.5 years, medication use; PE in weeks 2 and 4 of the treatment cycle, using the *EXCAP* program and *PointClickCare* APP	↑ Acceptance of the intervention for the preservation of functional capacity/habitual activity	↑ feasibility of the intervention, ↓ resistance to the use of technology
Kim et al. (2021) [19]; Republic of Korea	Open prospective single-arm study	Evaluating the effects of home PE with a personalized mobile APP on the amount of exercise, PA, QOL and depression in people with Parkinson’s disease	N = 28, average age ≥ 72 years; individualized multimodal PE for 8 weeks with personalized mobile APP	↑ Total exercise time, ↑ Functional components, ↑ BP, ↓ Depression	↑ EP adherence, ↑ QOL, ↓ Depression
Snoek et al. (2020) [16]; 5 European countries	Randomized clinical trial	Evaluate the effectiveness of a home-based rehabilitation program with remote monitoring and motivational coaching to achieve PE goals	N = 179, ≥65 years; Home cardiac rehabilitation for 6 months with smartphone (moderate PE, 5x/week, 30 min)	↑ VO_2_, ↓ Incidence of complications	↑ Motivation, Insurance, ↑ Membership
Bonato et al. (2024) [17]; Italy	Randomized clinical trial	Evaluating the effectiveness of a digital platform for prescribing home exercises for elderly people with sarcopenia, including people living with HIV	N = 188, ≥60 years or ≥50 years for PHIV+ sarcopenia; 48-week intervention using the Gym-Grow platform for prescribing home PE	83% adherence, ↑ Muscle strength, ↓ Difficulties finding the APP	↑ Efficiency in prescribing, ↑ Adherence

Legend: APP: Applications; N: Number; PE: Physical Exercise; QOL: Quality of Life; PA: Physical Activity; VO_2_: Maximum oxygen volume; PHIV: people living with HIV; HIV: Human Immunodeficiency Virus; EXCAP: Exercise for Cancer Patients. Symbols: ↑ = increase; ↓ = decrease; ≥: greater than or equal to.

## Data Availability

Not applicable.

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
