# Peer review of "Effects of the Prescription of Physical Exercises Mediated by Mobile Applications on the Health of Older Adults: A Systematic Review"

_geriatrics, 2025, doi:10.3390/geriatrics10050122_

Round 1

Reviewer 1 Report

Comments and Suggestions for Authors

This paper is a systematic review of the effects of prescribing physical exercises via mobile applications on the health of older adults.

This paper concludes that the above prescription positively affects older people's health.

This paper was written in accordance with systematic review methodology and added new insights.

This paper has a high degree of completeness in its description of methods.

This paper could be further improved by addressing the following 11 comments.

The major comment is the sixth comment.

Minor comments are all others.

1. Please insert a short sentence about the research gap immediately before “To analyze the effects” in line 17 of the abstract.

2. Please insert a short sentence describing the search strategy, including keywords, in the methods section of the abstract.

3. Please insert a short sentence about the screening criteria in the methods section of the abstract.

4. Please insert a short sentence about the data extraction process in the methods section of the abstract.

5. In lines 66-67 of the introduction section, please include the “C” element of “PICO” in the text.

P: older adults

I: MA-mediated PE prescription

C: Please describe the purpose of this study, including the “C” element.

O: health indicators

6. In line 87 of the Materials and Methods section, the following is stated:

“The following works were excluded: systematic reviews, meta-analyses, ・・”

The authors should not have excluded systematic reviews and meta-analyses that focused on MA-mediated PE prescription.

The authors may have excluded systematic reviews and meta-analyses that did *not* focus on MA-mediated PE prescription.

Please understand my concerns and revise the paper accordingly.

7. In the Materials and Methods section, please create a “Data synthesis and analysis” subsection and clearly state that you performed a “descriptive synthesis.”

8. In the Results section, between lines 172 and 173, please add a summary focusing on the outcomes regarding the characteristics of the included studies.

9. Please revise ML to MA in line 205 of the Discussion section.

10. In the Discussion section, lines 287 to 289, please include the limited number of databases used in the study among the limitations of the study.

11. Please move the description of future research in lines 296 to 300 of the Conclusions section to the Discussion section.

Please revise based on my comments. Alternatively, express your opposing opinion in response to my comments. The latter response is welcome.

Please indicate the line numbers of the revised sections and highlight them using red or yellow text backgrounds.

Author Response

First of all, we would like to thank you for your careful and thoughtful comments. We hope that the revisions meet your expectations. If you require any further clarification, please do not hesitate to contact us. All changes requested by the reviewers are highlighted in yellow.

 Comment 1: “Please insert a short sentence about the research gap immediately before “To analyze the effects” in line 17 of the abstract.”

Response: Thank you for your comment. In line 17, the following sentence was added: “However, there are still gaps in the effectiveness of prescribing physical exercise through mobile apps for older adults.”

Comment 2: “Please insert a short sentence describing the search strategy, including keywords, in the methods section of the abstract”.

Response: In lines 21 to 23, the following sentence was inserted: “The search strategy used the descriptors ‘elderly’, ‘physical exercise’, ‘mobile applications and ‘health’, combined with Boolean operators.”

Comment 3: “Please insert a short sentence about the screening criteria in the methods section of the abstract”.

Response: The following sentence was inserted in lines 23 to 27: “Screening followed previously defined eligibility criteria regarding population, intervention, outcomes, and study design”. Two independent reviewers extracted the data, mediated by a third party in case of disagreement; they selected and extracted data from the PubMed and VHL/Medline databases from 2004 to 2024.”

Comment 4: “Please insert a short sentence about the data extraction process in the methods section of the abstract.”

Response: The following sentence was inserted in lines 25 to 27: “Two independent reviewers extracted the data, with mediation by a third party in case of disagreement; they selected and extracted data from the PubMed and VHL/Medline databases from 2004 to 2024.”

Comment 5: “In lines 66-67 of the introduction section, please include the “C” element of “PICO” in the text”.

P: older adults

I: MA-mediated PE prescription

C: Please describe the purpose of this study, including the “C” element.

O: health indicators

Response: Thank you for your observation. Element C was included in the study objective, inserting the phrase in lines 73-74: “Thus, this study aimed to analyze the effects of prescribing physical exercises mediated by MA on the health indicators of elderly people, in comparison with practice without mediation or technological intervention.”

Comment 6: “In line 87 of the Materials and Methods section, the following is stated:
‘The following works were excluded: systematic reviews, meta-analyses.’
The authors should not have excluded systematic reviews and meta-analyses that focused on MA-mediated PE prescription. The authors may have excluded systematic reviews and meta-analyses that did not focus on MA-mediated PE prescription. Please understand my concerns and revise the paper accordingly
.”

Response: We appreciate this observation. The exclusion of systematic reviews and meta-analyses was limited to those that did not specifically address exercise prescription mediated by mobile applications. The focus of this review was on the analysis of primary studies with empirical data, in order to avoid overlapping information and to ensure methodological consistency. The text has been revised in lines 107 to 109 to clarify the justification for these exclusions. Thank you once again.

Comment 7: “In the Materials and Methods section, please create a “Data synthesis and analysis” subsection and clearly state that you performed a “descriptive synthesis.”.

 Response: Thank you for the recommendation. Section 2.1 Summary of Results has been added, including the corresponding text indicating the modifications that were implemented.

Comment 8: “In the Results section, between lines 172 and 173, please add a summary focused on the outcomes related to the characteristics of the included studies.”

Response: Thank you for this valuable suggestion. A summary focused on the outcomes related to the characteristics of the included studies has been added as requested, between lines 174 and 179”.

Comment 9: “Please revise ML to MA in line 205 of the Discussion section.”

Response: Thank you for pointing this out. The correction has been made.

Comment 10 : “In the Discussion section, lines 287 to 289, please include the limited number of databases used in the study among the limitations of the study”.

Response: Thank you very much for this observation. We acknowledge that the use of only PubMed and VHL/Medline represents a limitation, since the inclusion of additional databases could have broadened the scope of the search. Nevertheless, we chose these databases because they are widely recognized in the health field and provide consistent, though not exhaustive, coverage of the topic investigated. This limitation has been incorporated into the manuscript in lines 330 to 333: “An additional limitation was the search restricted to two databases (PubMed and VHL/Medline), which, although widely recognized and highly relevant in the health field, may have reduced the scope of the identified literature.”

Comment 11: Please move the description of future research from lines 296 to 300 of the Conclusions section to the Discussion section.”

Response: Thank you for this suggestion. The text has been moved and inserted in the Discussion section, between lines 337 and 341: “To consolidate this evidence, future research using more databases should evaluate the effectiveness of MA in different functional profiles for older adults, with long-term interventions, as well as explore factors such as digital literacy, accessibility, and technological support, which have not yet been addressed but are essential for the safe and effective implementation of these tools.”

Reviewer 2 Report

Comments and Suggestions for Authors

The Authors must see my remarks

(Extremely low sample - Why not and a "Meta-analysis"?????

Some Refs are missing

Author Response

First of all, we would like to thank you for your careful and thoughtful comments. We hope that the revisions meet your expectations. Should any further clarification be needed, we remain at your disposal. All changes requested by the reviewers are highlighted in yellow.

Comment 1: “Extremely small sample – Why not a meta-analysis?????”

Response: We appreciate this observation. Conducting a meta-analysis was not feasible due to the high methodological heterogeneity across the included studies (different populations, interventions, outcomes, and designs). For this reason, we opted for a descriptive synthesis, as recommended by the PRISMA protocol for situations in which sufficient statistical comparability is lacking.

Comment 2: “Add: ‘Materials and…’”

Response: Thank you for the correction. The phrase “Materials and Methods” has been added in line 19.

Comment 3: “Refs??? (74–79) …..”

Response: We appreciate this observation. We carefully reviewed the passage between lines 74 and 79 [after the textual changes, now lines 84 and 86] and confirmed that the references are already included and correspond to the statements presented. We maintained the citations in the text and revised the numbering to ensure clarity and consistency with the reference list.

Comment 4: “The reason(s)?????”

Response: We appreciate this observation. We have included in the text the justification for each exclusion criterion, clarifying that systematic reviews and meta-analyses without a specific focus on exercise prescription mediated by mobile applications were excluded because we prioritized primary studies, thereby avoiding overlapping results. We also detailed the reasons for excluding gray literature, studies outside the scope, works that only measured physical activity, and research involving other age groups. Please see lines 107–109.

Comment 5: “Extremely low sample….”

Response: We appreciate this observation. Indeed, the small sample sizes in the included studies represent a limitation that restricts the generalizability of the results. This point has been reinforced in the Limitations section, highlighting the need for future research with larger samples and more robust study designs.

Comment 6: “genders….”

Response: We appreciate the reviewer’s observation. The term has been revised in the text to ensure terminological consistency and alignment with international recommendations for good research practices. Please see line 254.

Comment 7: “Remove that paragraph….”

Response: We appreciate the suggestion. We recognize that the limitations mentioned are essential for the interpretation of the findings. Therefore, we chose to retain the content but revised the wording and relocated the paragraph to avoid redundancy, reinforcing these limitations only in the final section dedicated to this discussion. In this way, we ensured greater clarity, conciseness, and improved organization of the manuscript.

Reviewer 3 Report

Comments and Suggestions for Authors

I congratulate the authors for the interesting article. Find below my comments.

Abstract - generally comprehensive and well-structured.

Introduction - comprehensive and provides an interesting overview on the cururent state-of-the-art regarding the topic of interest. The review leads to a logical aim, which is very good.

I would suggest to add a final sentence (or two) after the study aim, where the authors would provide their hypothesis on the potential outcomes.

Methodology - well conducted with appropriate rigor. The study was registered in PROSPERO and the PICO strategy was followed.

I think the decision to include studies from only Pubmed and Medline, while not involving other databases like Scopus or WoS provides a significant limitation that needs to be addressed accordingly. The fact that many potentially relevant articles published in journals indexed in Scopus or WoS (and not Pubmed or Medline) might have been neglected presents in important point of discussion!

Figure 1 - "reports excluded" - we are aware that these reports were excluded due to not meeting the pre-set inclusion criteria. However, it would be interesting to describe the number of articles eliminated for every different inclusion criteria.

Results - well-described and understandable.

Discussion - is generally well-written and informative. Adequate articles are cited and the data are appropriately described and interpreted.

The limitation part needs to be restructured since a single sentence can not describe everything. This section should include reflective autocritique to your study, including the lack of specific plan on the kind/dosage of exercises to be included in the study. As so, now there is a rather big methodological heterogenity. The fact that there was a lack of control group in some studies should no be an issue to be passed by, but a rather methodological limitation when designing the inclusion/exclusion criteria. 

Author Response

First of all, we would like to thank you for your careful and thoughtful comments. We hope that the revisions meet your expectations. Should any further clarification be needed, we remain at your disposal. All changes requested by the reviewers are highlighted in yellow.

Comment 1: “Abstract – overall comprehensive and well structured.”

Response: We thank the reviewer for the positive comment regarding the Abstract.

Comment 2: “Introduction – comprehensive and provides an interesting overview of the current state of the art regarding the topic of interest. The review leads to a logical objective, which is very positive. I suggest adding one (or two) sentences at the end, after presenting the study objective, in which the authors state their hypothesis about the potential results.”

Response: We greatly appreciate this suggestion. At the end of the Introduction, we added a sentence stating our hypothesis that the use of mobile applications favors adherence and promotes positive effects on health indicators in older adults (lines 78–80).

Comment 3: “Methodology – well conducted and with appropriate rigor. The study was registered in PROSPERO and followed the PICO strategy. However, I believe that the decision to include only studies from PubMed and Medline, without involving other databases such as Scopus or WoS, represents a significant limitation that needs to be discussed. The fact that many potentially relevant articles published in journals indexed in Scopus or WoS (but not in PubMed or Medline) may have been overlooked constitutes an important point for discussion.”.

Response: Thank you very much for this observation. We acknowledge that the use of only PubMed and VHL/Medline represents a limitation, since the inclusion of additional databases could have broadened the scope of the search. Nevertheless, we opted for these databases because they are widely recognized in the health field and provide consistent, though not exhaustive, coverage of the investigated topic. This aspect has been incorporated as a limitation in the manuscript in lines 323–325: “An additional limitation was the search restricted to two databases (PubMed and VHL/Medline), which, although widely recognized and highly relevant in the health field, may have reduced the scope of the identified literature.”

Comment 4: “Figure 1 – ‘reports excluded’ – we know that these reports were excluded for not meeting the predefined inclusion criteria. However, it would be interesting to describe the number of articles eliminated for each specific inclusion criterion.”

Response: Thank you very much for this detail. The text has been revised to provide a description of the exclusion reasons, which are now detailed in lines 107 to 114.

Comment 5: “Results – well described and understandable.”

Response: We appreciate the positive comment. We are pleased to know that the Results section was considered clear and easy to understand.

Comment 6: “Discussion – overall well written and informative. Appropriate articles are cited, and the data are adequately described and interpreted.”

Response: We appreciate the positive comment. We are pleased to know that the Discussion section was considered well written.

Comment 6:“The Limitations section needs to be restructured, since a single sentence cannot describe all aspects. This section should include a reflective self-criticism of the study, including the absence of a specific plan regarding the type/dosage of exercises considered. This contributed to considerable methodological heterogeneity. Furthermore, the fact that some studies did not include a control group should not be minimized but rather acknowledged as a methodological limitation in the design of the inclusion/exclusion criteria.”

Response: We appreciate this important observation. The Limitations section has been critically restructured, and the changes have been incorporated in lines 326 to 340.

Round 2

Reviewer 1 Report

Comments and Suggestions for Authors

The author responded to all of my comments.
As a result, the paper improved.
It was already a well-developed paper at the first review stage.
Following the second review, I don't have any more comments.